# The determinants of nonprofit hospital CEO compensation

**Derek Jenkins**[1]*, **Marah N. Short**[1], **Vivian Ho**[1,2,3]

**1** Baker Institute for Public Policy, Rice University, Houston, Texas, United States of America, **2** Department of Economics, Rice University, Houston, Texas, United States of America, **3** Department of Medicine, Baylor College of Medicine, Houston, Texas, United States of America

* dj51@rice.edu

## Abstract

Hospital CEO salaries have grown quickly over the past two decades. We investigate correlates of rising nonprofit hospital CEO pay between 2012 and 2019 by merging compensation data from Candid's IRS 990 forms with hospital data from the National Academy for State Health Policy Hospital Cost Tool. Almost half of the measured increase in CEO compensation (44.5%) accrued to a "base case" CEO, who was leading a non-teaching hospital system or independent hospital with fewer than 100 beds that earned 0 profits and provided no charity care. Another 28.5% of the measured salary increase resulted from changes in the generosity with which observable metrics were rewarded, particularly the reward for heading a system with 500 or more beds. The remaining 27% resulted mostly from hospital systems or single hospitals that increased their profits or bed size over time. The increase in CEO compensation associated with leading larger healthcare systems and earning greater profits may explain the increase in healthcare system consolidation which has occurred over the last several years.

**Data Availability Statement:** The compensation data used in this study were originally collected by Candid by compiling information stored in IRS 990 tax forms that individually, are publicly available at (https://projects.propublica.org/nonprofits). Candid owns these data, and we cannot share these data

## Introduction

Hospital CEO salary has been growing quickly in the past two decades. Mean CEO compensation increased 92% from $1.6 million to $3.1 million among leaders of 22 major nonprofit medical centers between 2005 and 2015 [1]. While mean CEO pay by industry across the entire U.S. nonprofit sector ranged between only $100,000 and $200,000 in 2020, the mean pay of nonprofit hospital CEOs was around $620,000 [2]. In North Carolina, the nine largest health systems paid CEOs, chief financial officers, vice presidents and other highly paid administrative positions more than $1.75 billion from 2001–2021, with about 20 percent of this sum paid only to the CEOs of these health systems [3]. Not only is CEO compensation growing, but it is growing disproportionately relative to other healthcare workers; the wage gap between CEOs and mean salary of a registered nurse grew from 23:1 in 2005 to 44:1 in 2015 [1].

High levels of nonprofit CEO compensation may be warranted if the compensation is linked to better quality of care and greater community benefit. Joseph Newhouse first posited in 1970 that hospital trustees would reward nonprofit hospital CEOs for maximizing trustee

publicly per our data use agreement. Parties interested in purchasing these data should contact Candid at (https://Candid.org/use-our-data). Raw hospital financial data is available publicly for download at (https://tool.nashp.org). Our dataset is supplemented with the American Hospital Association survey data for system identification and teaching status. These data may not be shared publicly, but interested parties may inquire about purchases at (https://www.ahadata.com/aha-annual-survey-database). The authors had no special access privileges to the data in this study. Others are able to purchase and access the data in the same manner as the authors via the links provided in the data availability statement.

**Funding:** The authors declare funding support from Arnold Ventures grants No. 22-06846. The funders had no role in study design, data collection and analysis, publication decisions, or manuscript preparation.

**Competing interests:** I have read the journal's policy, and the authors of this manuscript have the following competing interests: the authors receive funding from Arnold Ventures. VH and MNS receive funding from the Health Care Service Corporation and the National Academy of State Health Policy. The funders had no role in study design, data collection and analysis, publication decision, or manuscript preparation. VH is a member of the Blue Cross Blue Shield of Texas Community Advisory Board, Community Health Choice: Houston Business Coalition on Healthcare Advisory Board, Texas Employers for Affordable Health Care Advisory Board, and the Blue Cross Blue Shield Research Alliance. VH receives support for attending membership committee meetings from the National Academy of Medicine. VH owned stock in IBM. "However, this does not alter our adherence to PLOS ONE policies on sharing data and materials."

utility instead of profits, by providing the optimal quantity and quality of care [4]. However, previous literature finds no association between better patient outcomes (i.e. risk adjusted mortality and readmissions) and executive compensation at nonprofit hospitals [5–7], although higher patient satisfaction has been linked to better pay. There have been mixed results on the association of CEO compensation with financial performance and a strong association with organization size [6,8,9].

Literature on agency theory suggests CEOs are incentivized to behave on behalf of the board of directors [10]. Previous empirical studies in a broader literature of CEOs in all industries have shown CEO compensation is associated with financial performance of the firm [11]. Hospital CEOs often have generous incentive packages encouraging them to meet various financial, managerial, and strategic goals. One news article series published in 2013 identified multiple large nonprofit health systems that linked bonuses to specified targets for profits, revenues, and hospital system growth [12]. In a 2017 survey of hospital CEOs, 69 percent reported that financial performance was a determinant of their incentive pay, while only 18 percent reported that serving the community, including activities to improve community or population health, affected their compensation [13]. Additionally, a 2010 study of nonprofit hospitals in Connecticut found that CEOs were driven to increase the number of privately insured patients as opposed to publicly insured ones [14]. These findings are consistent with the general economics literature, which suggests that CEO pay is closely linked to firm size and the size of the average firm in the economy [15,16].

As nonprofit hospitals consolidate, becoming more profitable, and amassing significant cash reserves, the share of expenses going to generating community benefit has not been increasing [17–19]. Federal laws obligate nonprofit hospitals to provide charity care and community benefits in order to preserve their tax-exempt status [17]. Our objective is to measure the determinants of executive compensation and estimate how the relationship between CEO compensation and operating profit, charity care, teaching status and hospital size has changed from 2012 to 2019.

## Methods

### Data sets

We obtained data on executive compensation from Candid's compilation of Internal Revenue Service (IRS) 990 tax forms filed by hospitals in 2012 and 2019 [20]. We focused on schedule H and the base 990 tax filings, which report executive compensation at the employer identification number (EIN) level. Some health systems file a group return using only one EIN, while others submit their tax filings at the hospital level and have multiple EINs.

Hospital financial data and characteristics were obtained from Medicare Cost Reports, which are maintained by the Center for Medicaid and Medicare Services. These cost reports are submitted annually by all hospitals treating Medicare patients. CMS publishes the information from hospitals' cost reports in its Healthcare Cost Report Information Systems (HCRIS) [21] database. We applied the methodology from the National Academy for State Health Policy's (NASHP's) Hospital Cost Tool to the HCRIS data to construct measures of hospital profits, spending on charity care, and cash reserve balances. Hospital profit is defined as net patient revenue minus total hospital operating costs. Charity care is defined as hospital operating costs for providing patient care under the hospital's charity care policy, less charity care patient payments and restricted grant funds received. We do not include bad debt in this amount. Charity care and profits are aggregated to the system level for health systems.

Following previous research by Joynt et. al. on nonprofit CEO pay, our sample includes CEOs of hospital systems and CEOs of single hospitals not affiliated with other hospitals [5].

We use the American Hospital Association annual survey to identify hospitals belonging to the same system, as well as teaching hospitals [22]. The AHA defines a teaching hospital as a hospital that has an American Medical Association residency program, is a member of the Council of Teaching Hospitals, or has a ratio of full-time equivalent interns and residents to beds of 0.25 or higher.

To merge the compensation data with our other data sources, we develop a CMS certification number (CCN) to EIN crosswalk (See S1.1 Fig in S1 Appendix) [23]. Since executive compensation is reported at the EIN level in the Candid personnel file, multiple hospitals belonging to the same system were in some cases merged with one EIN in the Candid data. In other cases, health systems reported to the IRS using multiple (hospital-level) EINs. There were 190 hospitals (93 in 2012 and 97 in 2019) in our data that were associated with EINs that included hospitals from different systems. These hospitals were dropped from our analysis, so that we could collapse the compensation data to the system level. In all cases we aggregated data (e.g. profits, number of beds, and charity care spending) to the system level. Our final sample consists of both hospital systems and independently operated hospitals that are not part of a larger hospital system. A detailed explanation of how the sample is constructed is available in S1.1 Fig in S1 Appendix [23].

## Identifying the CEO and CEO compensation

We identify a CEO for each hospital or health system using titles and compensation reported in the 990 forms. Health systems consisting of many hospitals may have multiple executives with titles such as, "CEO", "President", or "Executive Director". To tackle this issue, we begin with the methods of Song et al and identify executives with titles likely to indicate a CEO [24] and create a flag for executives likely to be the leader of the health system. After the possible leaders have been flagged, we designate the "CEO" of each system or independently operated hospital in our sample as the highest paid executive who is flagged as a leader for each organization. A deeper explanation of how we identified CEOs is available in S1.2 and S1.3 Figs in S1 Appendix. [23].

The measure of compensation used in this analysis is total compensation, which consists of W-2 and or 1099-Misc compensation including base compensation, bonus and incentive compensation, and other reportable compensation. Total compensation also includes retirement and other deferred compensation as well as nontaxable benefits.

## Empirical strategy

We first divided CEO compensation into deciles and compared hospital characteristics across these deciles. To understand the association between CEO compensation and hospital profit, charity care costs, teaching status and hospital size, we estimated a linear regression of the log of CEO wages on these independent variables. CEO wage, profit, and charity care costs are defined as total compensation, operating profits, and charity care spending for fiscal years 2012 and 2019. We constructed three categorical measures of hospital or system size. For systems in our sample, these measures reflect the total number of beds, discharges, or hospitals in the system. For our main specification, we created the following categories for number of beds, <100, 100–299, 300–499, and 500 or more beds. As a robustness check we also created categorical variables for number of adjusted discharges (<5,000, 5,000–19,999 20,000–99,999, >100,000), and number of hospitals in the system (single hospital, systems with 2–3 hospitals, systems with 4–9 hospitals, and systems with 10+ hospitals). Adjusted discharges scales up inpatient discharges to include outpatient care by multiplying inpatient discharges by the ratio of total hospital charges to inpatient charges. The coefficients represent the percentage change

**Table 1. Descriptive statistics by CEO compensation deciles.**

| | Bottom Decile | | Deciles 2–9 | | Top Decile | |
|---|---|---|---|---|---|---|
| *Variable* | **2012** | **2019** | **2012** | **2019** | **2012** | **2019** |
| | **Mean (SD)** | **Mean (SD)** | **Mean (SD)** | **Mean (SD)** | **Mean (SD)** | **Mean (SD)** |
| *CEO Wage* | 132,932 (34,118) | 148,546 (37,438) | 733,774 (481,157) | 883,027 (618,073) | 3,945,610 (1,926,787) | 5,624,209 (4,390,476) |
| *Profit ($ mill)* | 2.91 (8.37) | 1.79 (8.66) | 65.59 (129.19) | 88.23 (196.97) | 417.34 (505.57) | 788.27 (726.22) |
| *Charity Care ($ mill)* | 0.53 (1.39) | 0.39 (0.81) | 10.67 (26.70) | 10.84 (34.80) | 65.17 (95.50) | 84.35 (91.90) |
| *Cash Reserves ($ mill)* | 10.63 (22.42) | 9.25 (14.25) | 206.9 (383.9) | 338.5 (622.7) | 1528.9 (3053) | 2924.9 (3428) |
| *Teaching Status* | 0 (0) | 0 (0) | 0.06 (0.22) | 0.06 (0.21) | 0.36 (0.39) | 0.30 (0.30) |
| *Number of Beds* | 39.3(71.2) | 26.9 (25.9) | 274.3 (433.4) | 309.7 (638.2) | 1435.7 (1875.3) | 2258 (2319.6) |
| *Number of Hospitals* | 1.1 (0.4) | 1.0 (0.2) | 1.8 (2.7) | 2.1 (4.1) | 6.4 (10.8) | 10.4 (13.5) |
| *Adjusted Discharges* | 2,706 (4942) | 1,960 (3035) | 27,929 (41,448) | 34,890 (68,006) | 141,043 (179,758) | 247,444 (242,071) |

in CEO wages associated with a one-unit change in the independent variable. We present cash reserve levels in the descriptive statistics presented in Table 1, but are not able to include them in the regression analysis due to multicollinearity with the categorical variables for number of beds. S1.6 Table in S1 Appendix in the appendix presents a cross tabulation of the categorical variables for number of beds and number of hospitals [23].

The distribution of hospital profits is skewed to the right, because our sample contains both single hospitals and large health systems. While we are controlling for hospital size as a covariate, this skewness could still bias our regression estimates. Therefore, we estimated additional specifications where we transform profit using the cube root transformation that allows for negative values of profit.

We use a Oaxaca decomposition to quantify the portions of changes in CEO pay between 2012 and 2019 that are attributable to differences in hospital characteristics such as hospital size, versus increases in the generosity of compensation in 2019 versus 2012 for any given characteristic [25]. This decomposition has been used to decompose differences in wages between males and females that are attributable to differences in educational attainment or years of work experience by sex, versus differences in the rate at which these characteristics are rewarded for men versus women. The method has been applied to male-female wage differentials across time, industries, and countries [26]. [23] S1.5 Fig in S1 Appendix provides our regression specification, a summary of the Oaxaca method, and the regression coefficients and descriptive statistics used to construct the decomposition.

## Results

Our final sample consists of 1,113 independent hospitals and health systems in 2012 and 868 in 2019. The largest health system contained 101 hospitals in 2019. In 2012, CEOs of independent hospitals or health systems in our sample made an average of $996,000 (in 2019 dollars). By 2019, CEOs earned over 30 percent more, an average of $1.3 million. Over the same period, registered nurses' mean wages grew from $75,652 (in 2019 dollars) to $77,460 [27,28], only a 2.3% increase. Fig 1 graphs the levels of CEO compensation by year. As CEOs of independent and smaller hospital systems in 2012 consolidated into larger health systems by 2019, the distribution of CEO compensation shifted to the right.

Table 1 presents descriptive statistics on CEO compensation and hospital characteristics for 2012 and 2019. All descriptive statistics are reported in 2019 dollars. Total compensation grew for all decile groups, with the top decile experiencing a 42.5% increase, from $3.95 million in 2012 to $5.62 million in 2019. The middle eight, and bottom decile groups also had an increase in average CEO compensation of 20.3% and 11.7% respectively.

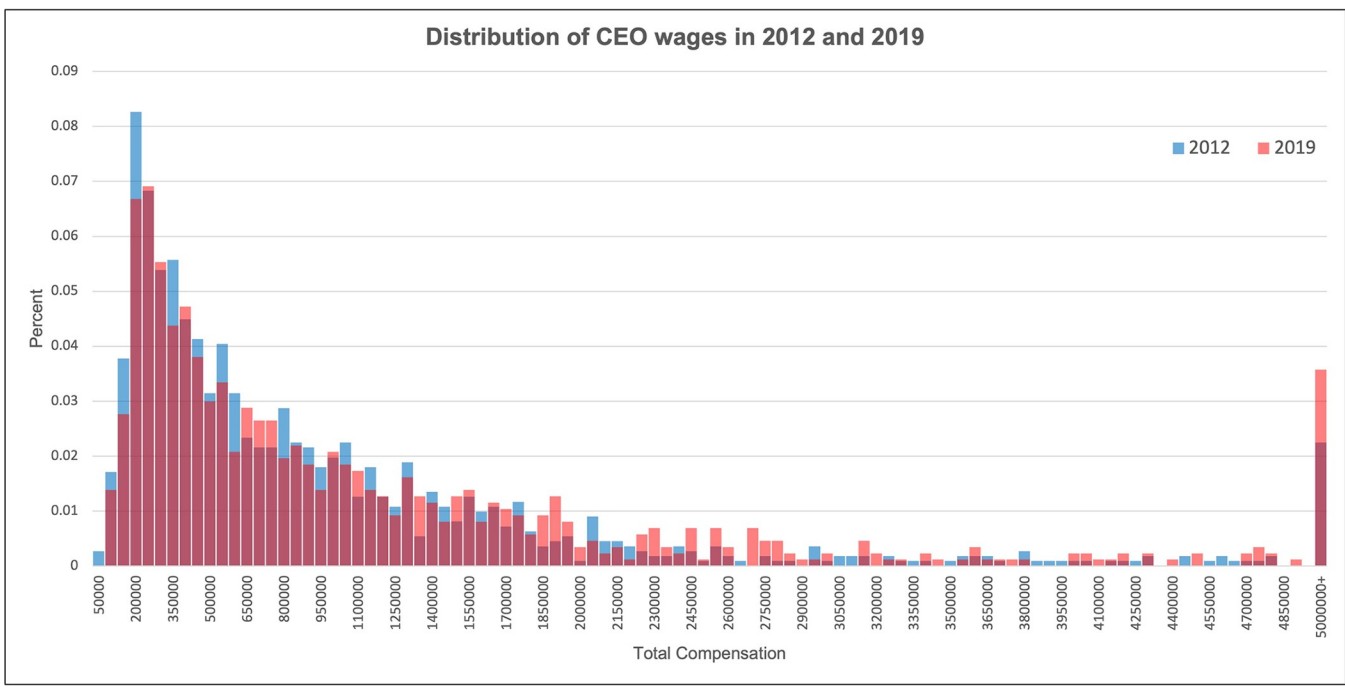

**Fig 1. Distribution of CEO wages in 2012 and 2019.** Fig 1 graphs the distributions of CEO compensation in 2012 and 2019. Compensation is presented in 2019 dollars and truncated at $5+ million.

Table 1 presents descriptive statistics for health systems and independent hospitals in our sample. CEO wage, profit, charity care, and cash reserves are all presented in 2019 dollars. The table is stratified by year and deciles of CEO compensation, with the middle 8 deciles presented together. CEO wage is the compensation of the highest paid executive, likely to be the organization leader. Number of beds, hospitals, and discharges are presented at the system level. Adjusted discharges are defined as the calculated inpatient and outpatient hospital discharges. This is computed by multiplying inpatient volume by an outpatient factor. The outpatient factor is defined as hospital charges divided by inpatient hospital charges. Teaching status is defined as the percentage of adjusted discharges that occurred at a teaching hospital within the hospital or health system.

Mean operating profits increased from $417.3 million to $788.3 million among health systems in the top decile of CEO compensation, while the profits of hospitals and health systems not in the top or bottom decile grew from $65.6 million to $88.2 million. Profits for hospitals in the bottom decile of CEO compensation fell from $2.91 to $1.79 million over the same time period.

Charity care increased among health systems in the top decile of CEO compensation, growing from $65.2 to $84.4 million, while the group containing the middle eight deciles of CEO compensation increased charity care from $10.7 to only $10.8 million. Charity care spending decreased from $0.53 to $0.39 million for the bottom decile.

## CEO compensation and hospital characteristics

A one million dollar increase in hospital profits was associated with a 0.06 percent (95% CI: 0.03, 0.09) increase in CEO pay in 2012, and a 0.05 percent (95% CI: 0.03,0.06) increase in CEO pay in 2019. A one million dollar increase in charity care had no significant association with CEO compensation.

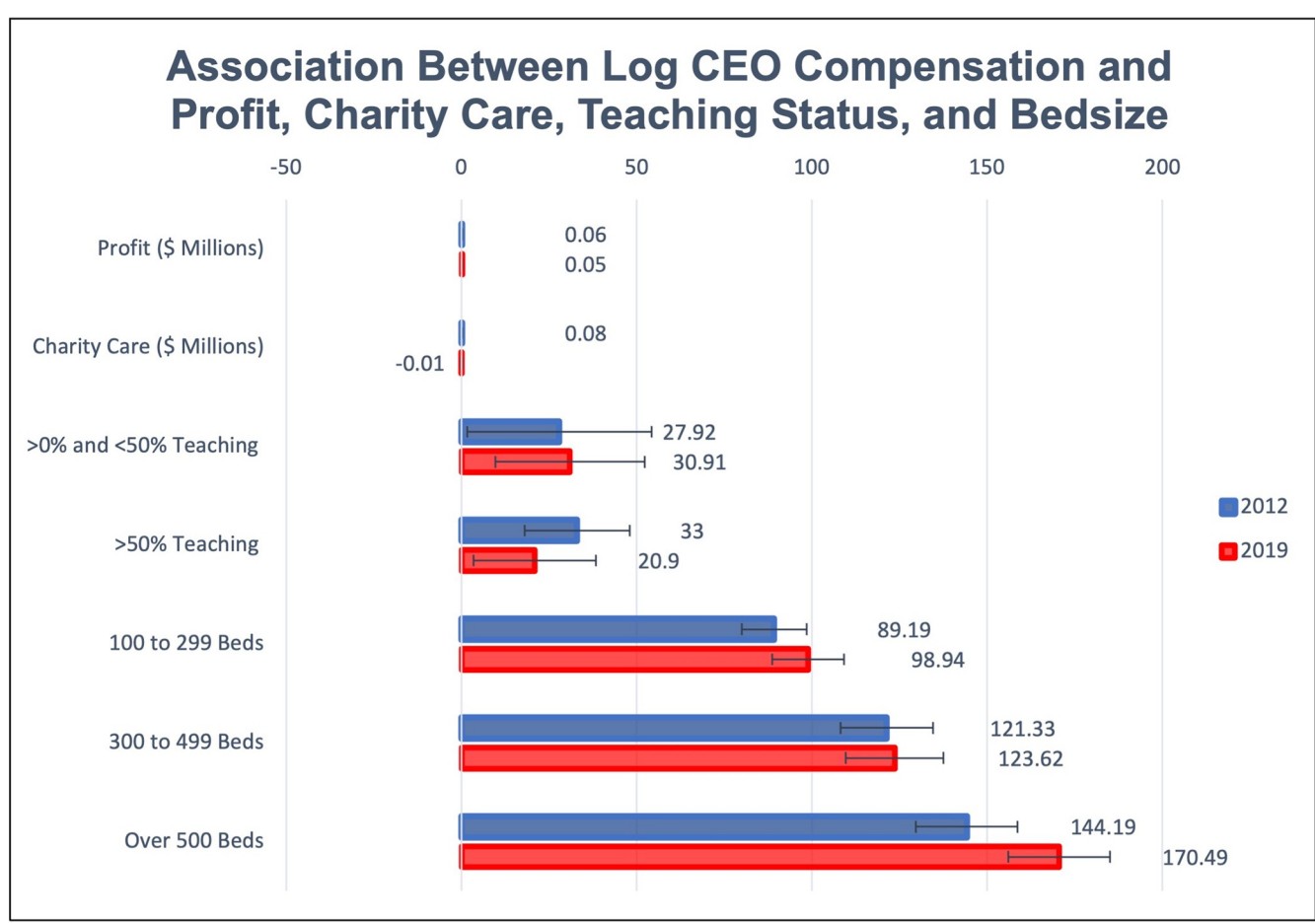

**Fig 2. Association between CEO compensation and profit, charity care, teaching status and Bedsize.** Fig 2 presents the results of our regression analyses for 2012 and 2019 where we regress profit, charity care, and categorical measures of teaching status and bed size on log CEO compensation. CEO compensation, profit (in $millions), and charity care (in $millions) are defined using 2019 dollars. Teaching status is defined as the percentage of adjusted discharges that occurred at a teaching hospital within the system or hospital. We multiply the regression coefficients by 100 to approximate the associated percentage change in CEO compensation. The estimates for profit and charity care can be interpreted as the estimated percentage change in CEO compensation associated with a one million dollar increase in profit or charity care. While the estimates for the bed size and teaching categorical variables should be interpreted as the increase in CEO compensation associated with each category of hospital size or teaching status relative to the reference group (hospitals with less than 100 beds, or hospitals/health systems with no adjusted discharges at teaching hospitals). Confidence intervals are presented at 95 percent confidence levels. The base case log CEO compensation (or the intercept term in our regression analysis, defined as estimated compensation for CEOs at hospitals with zero profits or charity care, less than 100 beds, and no adjusted discharges at teaching hospitals) is 12.56 in 2012 and 12.64 in 2019 which corresponds to 286,338 and 307,807 in 2019 dollars. The difference in base case CEO compensation from 2012 to 2019 is not statistically significant. There are 1,113 hospital or health systems in the 2012 sample and 868 in 2019.

Fig 2 presents the results from the regression analysis estimating the association between CEO compensation and profit, charity care spending and hospital size. Hospital size had a large, statistically significant association in both 2012 and 2019 with CEO compensation. In 2012, after controlling for profit, teaching status, and charity care, we estimate hospitals with 100–299 beds paid CEOs 89 percent more (95% CI: 80, 98) than the reference group of hospitals with fewer than 100 beds. Hospitals and health systems with 300–499 paid CEOs 121 (95% CI: 108, 135) percent more than the reference group, and hospitals with 500 or more paid their CEOs 144 (95% CI: 130, 159) percent more. By 2019 CEOs were paid 170 (95% CI: 156, 185) percent more than the reference group for leading a hospital with more than 500 beds, 124 (95% CI: 110, 138) percent more for hospitals with 300–499 beds, and 99 (95% CI: 89, 109)

percent more for hospitals with 100–299 beds. There was no increase in relative pay from 2012 to 2019 for the 300–499 beds category.

Although all of the coefficients (except on charity care) are significantly different from zero at the 95 percent confidence level, the individual coefficient estimates for 2019 versus 2012 are not significantly different from each other. However, Exhibit S1.7 in S1 Appendix presents the regression results of a sensitivity analysis including the interactions between a 2019 year dummy and the explanatory variables in our main regression. A Wald test shows the 2019 coefficients are jointly significantly different from the 2012 coefficients ($F_{(8, 1965)} = 3.63$; $P = 0.0012$) [23]. Additionally, the magnitude of the returns to managing a hospital or system with more beds was substantially larger in 2019 versus 2012. As a robustness check, we used discharges as the variable for hospital size, and the effects are similar. The results of the specification using a cube root transformation on profit are again similar and can be found in [23] Fig S1.4 in S1 Appendix.

The Oaxaca decomposition (See S.1.5 in S1 Appendix) of the change in log wages from 2012 to 2019 is presented in waterfall form in Fig 3 [23]. The results are similar when using the untransformed data, but we discuss the results in log form, which accounts for the skewness in CEO pay. The total difference in log wages is 0.162 from 2012 to 2019. The left-hand side of Fig 3 depicts the change in log wages resulting from differences in the magnitude of the coefficients for each characteristic in 2019 versus 2012 (i.e., the difference in compensation reward/penalty associated with a unit change in each hospital characteristic). By reporting the coefficient differentials sequentially, the waterfall format enables one to gauge the individual importance of each hospital/system characteristic in changing CEO compensation between 2012 and 2019. If the solid line is below (above) the bar attached to a hospital characteristic, then the

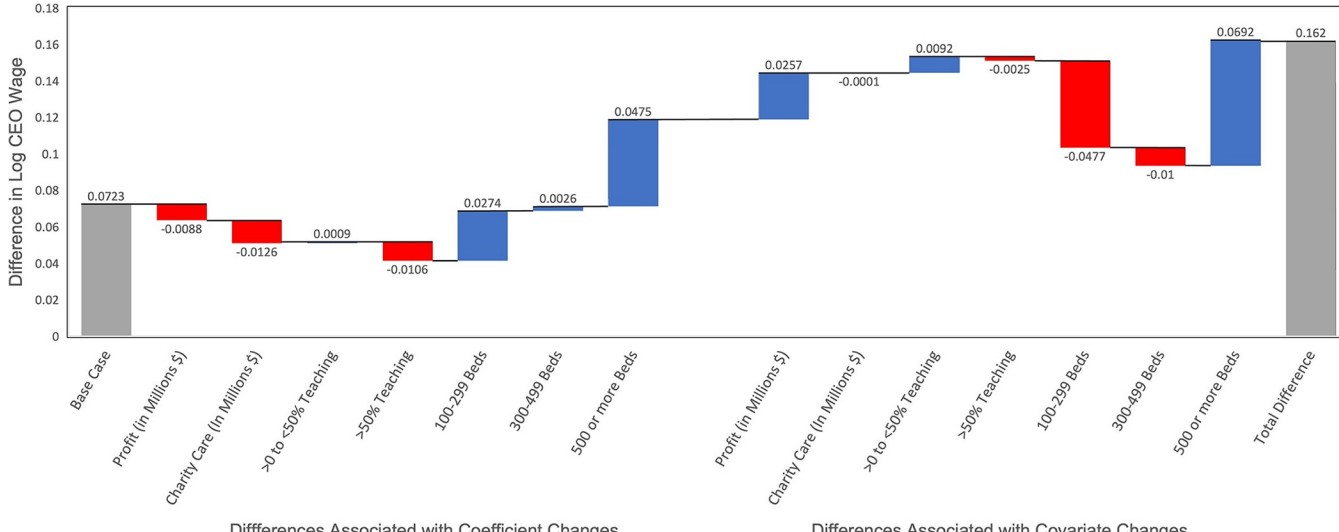

**Fig 3. Oaxaca decomposition.** Fig 3 presents the Oaxaca decomposition of the change in log wages from 2012 to 2019. The left-hand side of the chart presents the difference in log wages that is accounted for by the differences in coefficients from 2012 to 2019 and the right-hand side of presents the difference in log wages that is accounted for by the differences in the means of covariates. The leftmost bar represents the difference in regression intercept terms across years (the base case), which is the difference in log wage when the values of all explanatory variables equal 0. The right-most bar is the total difference in log wages, comprising the difference due to coefficient changes (including the base case), and the total of the difference due to covariate changes. Red (blue) columns indicate a negative (positive) contribution to the difference in log wages. Teaching status is defined as the percentage of adjusted discharges that occurred at a teaching hospital within the system or hospital.

difference in regression coefficients between 2012 and 2019 represents an increase (decline) in CEO compensation associated with that characteristic in 2019 versus 2012. Correspondingly, the right-hand side of the waterfall presents the sequential change in log wages that is accounted for by the differences in the mean of each hospital characteristic across time periods (e.g. how much the proportion of health systems with over 500 beds increased over time). The leftmost bar represents the difference in the base case across years, which is the difference in log wage for a hospital with fewer than 100 beds, no adjusted discharges at teaching hospitals, and zero profits and charity care in 2019 versus 2012.

On the left-hand side of Fig 3, the base case bar is by far the largest in magnitude, amounting to 0.072 relative to the overall log wage differential of 0.162. Compensation associated with being CEO of a larger hospital or health system was the next most important factor that was more generously rewarded in 2019 versus 2012. The magnitude of the bar for heading a hospital or system with 100 to 299 beds was 0.027, and the bar for heading a system with over 500 beds was even larger (0.048). In contrast, payments for greater profits, teaching status, or charity care did not differ markedly between 2012 and 2019.

On the right side of Fig 3, the larger share of CEOs managing systems with over 500 beds in 2019 versus 2012 was associated with 0.069 greater log wages. Because hospitals consolidated between 2012 and 2019, much of this differential was offset by a 0.048 reduction in log wages associated with fewer CEOs heading systems with 100 to 299 beds. Mean profits adjusted for inflation also rose between 2012 and 2019, amounting to 0.026 greater log wages. [23] S1.5 Fig in S1 Appendix presents the coefficients and means used to construct these estimates.

Overall, 28.5% of the difference in log wages is explained by coefficient differences from 2019 and 2012 and 27% of the difference is explained by changes in hospital characteristics across time. The remaining 44.5% reflects the difference in log wages for the base case hospital in 2019 versus 2012.

## Discussion

### Policy implications

In an analysis of 1,113 nonprofit health systems and independent hospitals in 2012 and 868 in 2019, we found that mean CEO compensation rose 30% between 2012 and 2019, but mean salary for CEOs in the top decile grew even more, by 42%. Similar to other studies that apply the Oaxaca decomposition to determine the relative importance of differences in characteristics versus differences in payment generosity for those characteristics, we perform our analysis using log wages to control for the observed skewness in earnings. While changes in log wages may be difficult to conceptualize, the method illustrates that almost half (44.5%) of the increase in log CEO compensation between 2019 and 2012 was attributable to an increase in pay for the base case hospital. This result suggests that a hospital CEO that did not improve her performance over time still enjoyed a significant (inflation-adjusted) boost in pay. Part of this pay increase might be explained by the increased complexity of operating a hospital or system, including a wider range of affiliated outpatient facilities such as ambulatory surgery centers, and greater vertical integration with physicians, which we did not account for in our analysis.

Health systems or hospitals that were larger in size or earned higher profits than the base case hospital accounted for most of the additional 27% increase in CEO pay observed over the time period. Of the remaining 28.5% increase in log CEO compensation, most was attributable to an increase in pay generosity for CEOs managing systems with over 500 beds.

The substantial increase in compensation that occurred for all nonprofit hospital CEOs regardless of performance of their institution is consistent with reports from other sources of a widening gap between healthcare executive pay versus nurses' compensation [1]. The increase

in CEO compensation associated with leading larger healthcare systems and earning greater profits may explain the massive increase in healthcare system consolidation which has occurred over the last several years.

A previous study of nonprofit hospital CEO compensation by Joynt et. al. [5] and another by Mulligan et. al. [6] did not report an association between either profits or system size and compensation, although both studies found that greater patient satisfaction was linked to higher CEO pay. Both studies specified profits in terms of margins (the ratio of net income to revenues), while we specified profits in millions of dollars. We specified profits in dollar terms, because this criterion has been used by both for-profit and nonprofit hospital boards to determine incentive pay [12,29]. Both studies included number of beds as an explanatory variable in their regressions, although neither study reported coefficient estimates for this hospital characteristic. Mulligan et. al. chose to exclude CEOs of hospital systems from their sample and instead included CEOs of individual hospitals within systems. The range of profits and facility size in their sample would be more limited relative to our analysis or that in Joynt et. al, which may partially explain a lack of association between CEO pay and hospital profits or size.

Our analysis advances the literature by quantifying the significant role that higher profits and increasing system size play in explaining rising CEO pay. Unlike the two studies just mentioned, we did not measure the association between hospital quality and CEO pay. However, both past studies found no association between CEO pay and a range of process of care and patient outcome measures. Both past studies found a link between greater patient satisfaction and higher CEO pay. However, the mean patient recommendation of hospitals in Mulligan et. al's analysis rose from 68.8 in 2009 to only 71.2 (out of 100) in 2014, which, when applied to the study's coefficient estimates, implies an almost negligible increase in CEO pay [6].

Recently, there have been news articles highlighting state and local efforts to either cap CEO compensation or remove tax exempt status for hospitals. In Pennsylvania, public school districts pushed back against the nonprofit health system in the area "Tower Health," and four hospitals had their tax-exempt status revoked. The executive salaries at these four locations were deemed to be excessively large, and none of these four hospitals were spending even one percent of their expenses on charity care [7]. This case is not the only example of state and local government pushing back against nonprofit hospital leaders prioritizing executive compensation over community benefit. Recently, in California, Oregon, and Arizona the Service Employees International Union (SEIU) have proposed capping hospital executive salaries considering reports showing how large these salaries have grown [30]. Affordable healthcare continues to be a problem for many Americans. As many as 41% of adults currently have debt caused by medical or dental bills [31]. Misaligned incentives for executives at nonprofit hospitals contributes to the rising burden of healthcare costs to American families and suggests their financial priorities are not aligned with the obligations to the communities they serve. If there is little distinction between nonprofit and for-profit hospital behavior [17,18], additional policies to ensure nonprofit hospitals deliver community benefits should be considered, such as reporting of performance criteria that determine CEO bonuses and required reporting of hospital CEO to employee pay ratios.

## Limitations

CEOs are incentivized by the board of directors to meet various organizational goals. Thus, decisions on CEO compensation are likely endogenous, and our estimates of the association between CEO compensation and hospital performance cannot be interpreted as causal effects. However, our decomposition provides a useful descriptive analysis of what financial and organizational characteristics are associated with rising CEO pay.

Some hospitals did not include a CEO in their 990 forms. Many 990 forms used other titles for the CEO (i.e. President/COO, Executive Officer, Chairman/President), while some others had missing titles. In addition, some executives are listed as a CEO at one hospital and hold a different position at a related hospital. These issues are discussed in greater detail in [23] S1.2 Fig in S1 Appendix. In some cases, these challenges could lead us to selecting a higher paid executive within the health system who is not the current CEO. Additionally, individual characteristics such as age, sex, and tenure may be important factors in determining CEO pay, but we do not observe these in the data.

Another limitation of our analysis is that data for the IRS 990 forms can be reported for either the calendar year or the fiscal year. Hospital cost reports were analyzed for the fiscal years 2012 and 2019. For this reason, we do not conduct a sensitivity analysis using lagged values of profits and charity care, because it is likely that many of our observations already represent lagged values in our sample.

We may have underestimated profits for some healthcare systems, which are reported at the hospital level in the HCT dataset. When we aggregated hospitals from the NASHP HCT to the health system level, some hospitals were dropped from the analysis, because we could not find a matching facility in the Candid dataset. However, 91.9% of the nonprofit hospitals in the HCT data were matched to the final data set in 2012 and in 2019 94.3% were matched, so this shortfall in profits is likely small. Additionally, profits can be volatile, and because we only estimate cross-section regressions of observables associated with CEO compensation in 2012 and 2019, our coefficient estimates on profit may be biased toward zero.

## Conclusion

Executive compensation grew substantially for all CEOs of nonprofit hospitals and healthcare systems from 2012 to 2019. Compensation increases for CEOs of healthcare systems with over 500 beds were even larger, and the proportion of CEOs leading such large systems grew over time. Higher profits were the next most important determinant of higher CEO pay, while changes in charity care and their association with CEO compensation were negligible. Rising executive compensation is contributing to the affordability crisis in American healthcare and should remain in the forefront of the minds of policy makers.

## Supporting information

**S1 Appendix. Supplement to the determinants of nonprofit hospital CEO compensation.** (DOCX)

## Acknowledgments

The authors thank Marilyn Bartlett for helping them understand the National Academy for State Health Policy Hospital Cost Tool; and Mireille Jacobson for her insightful comments.

## Author Contributions

**Data curation:** Derek Jenkins, Marah N. Short, Vivian Ho.

**Formal analysis:** Derek Jenkins, Marah N. Short, Vivian Ho.

**Funding acquisition:** Vivian Ho.

**Methodology:** Derek Jenkins, Marah N. Short, Vivian Ho.

**Project administration:** Vivian Ho.

**Supervision:** Vivian Ho.

**Writing – original draft:** Derek Jenkins.

**Writing – review & editing:** Marah N. Short, Vivian Ho.

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
