## [Decision Letter · Decision Letter 0]

8 Apr 2024

PONE-D-24-07430The determinants of nonprofit hospital CEO compensationPLOS ONE

Dear Dr. Jenkins,

Thank you for submitting your manuscript to PLOS ONE. After careful consideration, we feel that it has merit but does not fully meet PLOS ONE’s publication criteria as it currently stands. Therefore, we invite you to submit a revised version of the manuscript that addresses the points raised during the review process. In particular, we would like you to provide a more precise description of the empirical analysis, including more precise definitions and an equation for the specification estimated. Also, endogeneity is a significant issue. The authors carefully use the term "association" but do not explore how these associations arise. 

We look forward to receiving your revised manuscript.

Kind regards,

Gabriel A. Picone

Academic Editor

PLOS ONE

Journal Requirements:

   "I have read the journal's policy and the authors of this manuscript have the following competing interests: the authors receive funding from Arnold Ventures. VH and MNS receive funding from the Heath Care Service Corporation and National Academy of State Health Policy.  The funders had no role in study design, data collection and analysis, decision to publish, or preparation of the manuscript.

VH is a member of the Blue Cross Blue Shield of Texas Community Advisory Board; Community Health Choice: Houston Business Coalition on Healthcare Advisory Board; Texas Employers for Affordable Health Care Advisory Board and the Blue Cross Blue Shield Research Alliance. VH receives support for attending membership committee meetings from the National Academy of Medicine. VH owned stock in IBM. 

"

We note that you received funding from a commercial source:  Arnold Ventures, Heath Care Service Corporation and National Academy of State Health Policy

Reviewers' comments:

Reviewer's Responses to Questions

**Comments to the Author**

1. Is the manuscript technically sound, and do the data support the conclusions?

Reviewer #1: Partly

2. Has the statistical analysis been performed appropriately and rigorously? 

Reviewer #1: Yes

3. Have the authors made all data underlying the findings in their manuscript fully available?

Reviewer #1: No

4. Is the manuscript presented in an intelligible fashion and written in standard English?

Reviewer #1: Yes

5. Review Comments to the Author

Reviewer #1: General Comments

1. The topic of the paper is an interesting and an important one. The paper potentially contributes to the literature on determinants of CEO compensation (although the existing economic literature on the topic is not reviewed. There is one citation (13) from this literature.

2. On page 3, lines 69-70, the authors indicate that hospital CEOs are offered generous incentive pages to meet financial, managerial, and strategic goals. If so, would not such covariates as profit, cash reserves and hospital and and system characteristics be endogenous? A survey of hospital CEOs is cited on lines 72 ff, the results indicating that achieving financial goals is a major determinant of incentive pay. Then profit and CEO relationship is certainly bidirectional. Although decompensation studies do not typically deal with endogeneity, the case for doing so seems strong here.

3. Teaching status is never defined. There are alternative definitions, including presence of an accredited residency program, the number of residents per bed, membership in the American Association of Medical Colleges.

4. There are other cases for which details about empirical specification is lacking. I assume the authors picked the highest paid CEO when there are several CEOs in the hospital system, but this is not stated. Is the observational unit the system when the hospital is part of a system and a hospital otherwise? Are there systems in which the total number of beds is say 250? Or do most systems have 500+ beds. Some systems, e.g., HCA have many multiples of 500 beds. Is profit defined for one year, the current year or a lagged year, or the year the CEO was appointed? It would be good to see the main equation specification that was estimated.

5. There is no theoretical framework. Changes in coefficients between the two years could reflect structural changes in markets for hospital executives. Demand for such executives could have shifted outward relative to a stable inelastic supply curve. How would one know whether ‘’executive compensation is excessive.”? (line 396). This could explain the base case change in coefficients. Journalists will be looking for a statement that compensation is excessive, but the paper does not present empirical evidence for this. The price of eggs may rise rapidly because of a supply shock. This does not mean that the price of eggs is “excessive.”

6. Will readers understand the purpose of and how an Oaxaca decomposition is performed? I am among potential interested readers who does not know what a waterfall form is.

7. What happened to the asset covariate?

Specific Comments (the numbers are line numbers)

52. Use mean rather than average.

53. The $1.75 billion figure is hard to understand. Divide by 9?

55. What is a “handful?”

61 ff. More precisely, in the Newhouse model, the trustees have a utility function which depends on quantity and quality. Optimal quantity and quality are determined by maximizing trustee utility subject to a zero- profit constraint.

64. Which patient outcomes have been studied. Hospital quality is hard to measure from a distance. And there are many quality attributes.

78. One study I have read finds that payer mix does not change after hospital mergers.

86. What do you mean by charity care?. There is uncompensated care which is the sum of charity care (care the hospital expects to subsidize at patient admission) and bad debt care (revenue the hospitals expect to collect but does not). Uncompensated care is a more comprehensive definition.

98. How do you deal with the EIN problem?

129. Then hospitals in systems have multiple CEOs. Chose the highest paid CEO? I checked a chain I am familiar with. The head of a hospital that is part of the chain but not the leading hospital has a “President.”

144. Be more specific about how hospital bed size is defined in the case of systems.

156. You know from your data that profits are skewed to the right and that there are negative values. This is more than “likely.”

160. What does “back transform” mean?

165 ff. More explanation needed.

Table 1. You mention cash reserves in the table. But they disappear elsewhere. Cash reserves can be highly endogenous in that a highly paid entrepreneur may use such funds for hospital/system improvement/expansion. Why use cash reserves and not other items from the balance sheet?

223. A 0.06% change is trivial. Also, the 95%Cis overlap for all results in Fig. 2 (see bars and whiskers), implying no statistically significant changes in structure between 2012 and 2019.

247. The definition of teaching status must include a definition of teaching hospital.

286.larger hospital or health system. Again, we need a precise definition of how size is constructed. Does system trump hospital when a hospital belongs to a system?

295. Are there systems with 100-299 beds? If so, what is the N for this category?

394. This statement may reflect equity considerations. But relative wages can change over time for many reasons. There is a literature on monopsony in a hospital context. Is this being implied here?

378. Some of the points I make under General Comments apply here. In particular, it is likely that some hospital executives receive generous pay offers because boards think the execs can help solve the hospital’s financial problems.

388. Profits are volatile. For this reason, the parameter estimate may be biased toward zero (consider it an error in variable).

Fig. 1. How do you handle the open-ended category at the far right in the regression analysis?

Fig. 2. I prefer regression results for the preferred specifications. But perhaps this is a matter of taste. I have a stronger preference for discussing methods in the text. None of the parameter estimates are statistically different between 2012 and 2019. What is the sample size? I do not see it here or elsewhere in the paper. Hospitals dropped?

Fig. 3. Can there be better titles than unexplained coefficient effects? Effects implies causation.

They are really changes in how CEO time is priced. The right side is changes in the mean values of pay determinants. Again, methods should be explained in the text.

6. PLOS authors have the option to publish the peer review history of their article (what does this mean?). If published, this will include your full peer review and any attached files.

Reviewer #1: No

---

## [Author Response · Author response to Decision Letter 0]

30 May 2024

All line number references below refer to the clean version of the manuscript. 

Reviewer #1: General Comments

1. The topic of the paper is an interesting and an important one. The paper potentially contributes to the literature on determinants of CEO compensation (although the existing economic literature on the topic is not reviewed. There is one citation (13) from this literature.

Thank you for the suggestion that we include more sources from the broader economics literature on CEO compensation. On line 71 we have added the text below, along with citations of (Mirrlees Bell J Econ 1976 [10] and Murphy Handbook of the Economics of Finance 2013 [11]), 

“Literature on agency theory suggests CEOs are incentivized to behave on behalf of the board of directors [10]. Previous empirical studies in a broader literature of CEOs in all industries have shown CEO compensation is associated with financial performance of the firm. [11]”

2. On page 3, lines 69-70, the authors indicate that hospital CEOs are offered generous incentive pages to meet financial, managerial, and strategic goals. If so, would not such covariates as profit, cash reserves and hospital and system characteristics be endogenous? A survey of hospital CEOs is cited on lines 72 ff, the results indicating that achieving financial goals is a major determinant of incentive pay. Then profit and CEO relationship is certainly bidirectional. Although decompensation studies do not typically deal with endogeneity, the case for doing so seems strong here.

We agree that CEO compensation is likely endogenous and we do not attempt to estimate causal effects in this paper. Thank you for letting us know that this was not clear to the reader. We have added the following sentences on line 402 at the beginning of the limitations section to acknowledge that we are not estimating a causal relationship. 

“CEOs are incentivized by the board of directors to meet various organizational goals. Thus, decisions on CEO compensation are likely endogenous, and our estimates of the association between CEO compensation and hospital performance cannot be interpreted as causal effects. However, our decomposition provides a useful descriptive analysis of what financial and organizational characteristics are associated with rising CEO pay.

3. Teaching status is never defined. There are alternative definitions, including presence of an accredited residency program, the number of residents per bed, membership in the American Association of Medical Colleges.

We apologize for the oversight. The following definition of teaching status has been added on line 117. 

“The AHA defines a teaching hospital as a hospital that has an American Medical Association residency program, is a member of the Council of Teaching Hospitals, or has a ratio of full-time equivalent interns and residents to beds of 0.25 or higher.”

4. There are other cases for which details about empirical specification is lacking. I assume the authors picked the highest paid CEO when there are several CEOs in the hospital system, but this is not stated. Is the observational unit the system when the hospital is part of a system and a hospital otherwise? Are there systems in which the total number of beds is say 250? Or do most systems have 500+ beds. Some systems, e.g., HCA have many multiples of 500 beds. Is profit defined for one year, the current year or a lagged year, or the year the CEO was appointed? It would be good to see the main equation specification that was estimated.

• Thank you for letting us know that these details were not clear in the main text. All hospitals within systems are aggregated to the system level. Our final sample includes both hospital systems, and independently operated hospitals. The following sentence has been added to line 128. 

“In all cases we aggregated data (e.g. profits, number of beds, and charity care spending) to the system level. Our final sample consists of both hospital systems and independently operated hospitals that are not part of a larger hospital system.”

• We have also updated the explanation of CEO identification in the main text to be more specific. The sentence beginning on line 137 now reads, 

“After the possible leaders have been flagged, we designate the “CEO” of each system or independently operated hospital in our sample as the highest paid executive who is flagged as a leader for each organization. A deeper explanation of how we identified CEOs is available in appendix Figs S1.2 and S1.3.[23].”

In the appendix, the details of CEO identification are explained at length. We felt it was cumbersome to include the full discussion of these methods in the main text. 

• Thank you for this question. Yes, there are systems where the total numbers of beds is lower than 250. We have added a cross tabulation to the appendix for our categorical variables of bed size and number of hospitals. We include a mention to this table on line 164 with the sentence, 

• 

“Table S1.6 in the appendix presents a cross tabulation of the categorical variables for number of beds and number of hospitals. [23]”

• Profit is defined as total operating profit in the current fiscal year. We have added a sentence in the text on line 151 that reads, 

“CEO wage, profit, and charity care costs are defined as total compensation, operating profits, and charity care spending for fiscal years 2012 and 2019.”

Some other studies use lagged values of financial performance, which we could not do with confidence with our merged dataset. We have added an explanation on line 417 in the Limitations section:

“Another limitation of our analysis is that data for the IRS 990 forms can be reported for either the calendar year or the fiscal year. Hospital cost reports were analyzed for the fiscal years 2012 and 2019. For this reason, we do not conduct a sensitivity analysis using lagged values of profits and charity care, because it is likely that many of our observations already represent lagged values in our sample.”

• In the last sentence before the Results section, we have added a mention that our regression specification is included in the Appendix Fig S.1.5.

5. There is no theoretical framework. Changes in coefficients between the two years could reflect structural changes in markets for hospital executives. Demand for such executives could have shifted outward relative to a stable inelastic supply curve. How would one know whether ‘’executive compensation is excessive.”? (line 396). This could explain the base case change in coefficients. Journalists will be looking for a statement that compensation is excessive, but the paper does not present empirical evidence for this. The price of eggs may rise rapidly because of a supply shock. This does not mean that the price of eggs is “excessive.”

You are correct that we do not present a theoretical framework and cannot claim from our analysis that executive compensation is excessive. We agree the statement you refer to was misleading. The paragraph on page 18 has been revised to remove this sentence. We also revised the sentence beginning on line 393 to remove the phrase “excessive compensation” and now reads, 

“Misaligned incentives for executives at nonprofit hospitals contributes to the rising burden of healthcare costs to American families and suggests their financial priorities are not aligned with the obligations to the communities they serve”

6. Will readers understand the purpose of and how an Oaxaca decomposition is performed? I am among potential interested readers who does not know what a waterfall form is.

Thank you for pointing out this shortcoming. We have expanded the explanation of the Oaxaca decomposition on page 8 line 174 to read:

“We use a Oaxaca decomposition to quantify the portions of changes in CEO pay between 2012 and 2019 that are attributable to differences in hospital characteristics such as hospital size, versus increases in the generosity of compensation in 2019 versus 2012 for any given characteristic.[25] This decomposition has been used to decompose differences in wages between males and females that are attributable to differences in educational attainment or years of work experience by sex, versus differences in the rate at which these characteristics are rewarded for men versus women. The method has been applied to male-female wage differentials across time, industries, and countries.[26] Appendix[23] Fig S1.5 provides a summary of the Oaxaca method and the regression coefficients and descriptive statistics used to construct the decomposition.”

The concept of graphing results in waterfall format is more general than a Oaxaca decomposition. We have modified the text on page 13 line 281 to clarify this point. Moreover, we modified Fig 3, so that it is easier to see the bar associated with each hospital characteristic, so that one can readily tell whether each factor contributes to a positive or negative change in CEO pay in 2019 vs 2012:

The Oaxaca decomposition (See Appendix S.1.5) of the change in log wages from 2012 to 2019 is presented in waterfall form in Fig 3… The left-hand side of Fig 3 depicts the change in log wages resulting from differences in the magnitude of the coefficients for each characteristic in 2019 versus 2012 (i.e., the difference in compensation reward/penalty associated with a unit change in each hospital characteristic). By reporting the coefficient differentials sequentially, the waterfall format enables one to gauge the individual importance of each hospital/system characteristic in changing CEO compensation between 2012 and 2019. If the solid line is below (above) the bar attached to a hospital characteristic, then the difference in regression coefficients between 2012 and 2019 represents an increase (decline) in CEO compensation associated with that characteristic in 2019 versus 2012. Correspondingly, the right-hand side of the waterfall presents the sequential change in log wages that is accounted for by the differences in the mean of each hospital characteristic across time periods (e.g. how much the proportion of health systems with over 500 beds increased over time). The leftmost bar represents the difference in the base case across years, which is the difference in log wage for a hospital with fewer than 100 beds, no adjusted discharges at teaching hospitals, and zero profits and charity care in 2019 versus 2012.

7. What happened to the asset covariate?

The cash reserves variable was highly correlated with bed size and could not be included in the regression. We felt it still provided useful context, so we kept it in Table 1 as a descriptive statistic. A sentence has been added to the methods section on line 162 stating, 

“We present cash reserve levels in the descriptive statistics presented in Table 1, but are not able to include them in the regression analysis due to multicollinearity with the categorical variables for number of beds.”

Specific Comments (the numbers are line numbers)

52. Use mean rather than average.

Thank you for this correction. On line 50 “average” has been changed to “mean” in the text. 

53. The $1.75 billion figure is hard to understand. Divide by 9?

This figure refers to the total amount paid to highly paid executives, which includes chief financial officers, vice presidents, chief legal officers and management positions over clinical care staff like chief medical officers. It should be interpreted as spending on highly paid administrative staff, not involved with clinical care. We have rephrased the sentence on line 54 to read: 

“In North Carolina, the nine largest health systems paid CEOs, chief financial officers, vice presidents and other highly paid administrative positions more than $1.75 billion from 2001-2021…”

55. What is a “handful?”

We are referring to the nine hospital CEOs at the health systems mentioned earlier in the sentence. We have revised the text on line 54 to read, 

“In North Carolina, the nine largest health systems paid CEOs, chief financial officers, vice presidents and other highly paid administrative positions more than $1.75 billion from 2001-2021, with about 20 percent of this sum paid only to the CEOs of these health systems.”

61 ff. More precisely, in the Newhouse model, the trustees have a utility function which depends on quantity and quality. Optimal quantity and quality are determined by maximizing trustee utility subject to a zero- profit constraint.

Thank you for this correction. We have revised the sentence on line 62 to read, “Joseph Newhouse first posited in 1970 that hospital trustees would reward nonprofit hospital CEOs for maximizing trustee utility instead of profits, by providing the optimal quantity and quality of care.”

64. Which patient outcomes have been studied. Hospital quality is hard to measure from a distance. And there are many quality attributes.

The literature we reference includes risk adjusted mortality and readmissions as patient outcomes. In the main text we have updated the sentence on line 64 to read, “However, previous literature finds no association between better patient outcomes (i.e. risk adjusted mortality and readmissions)”. 

78. One study I have read finds that payer mix does not change after hospital mergers.

Thank you for this comment. We have updated the sentence on line 80 to read, 

“Additionally, a 2010 study of nonprofit hospitals in Connecticut found that CEOs were driven to increase the number of privately insured patients as opposed to publicly insured ones.[12]

86. What do you mean by charity care?. There is uncompensated care which is the sum of charity care (care the hospital expects to subsidize at patient admission) and bad debt care (revenue the hospitals expect to collect but does not). Uncompensated care is a more comprehensive definition.

We apologize for the oversight. We have revised the text on line 109 to provide a better definition. 

“Charity care is defined as hospital operating costs for providing patient care under the hospital's charity care policy, less charity care patient payments and restricted grant funds received. We do not include bad debt in this amount. Charity care and profits are aggregated to the system level for health systems.”

98. How do you deal with the EIN problem?

Thank you for letting us know this wasn’t clear. Our treatment of EINs attached to a set of hospitals belonging to two or more systems is explained in detail in the appendix exhibit S1.1, but we have added the following two sentences on line 125,

 “There were 190 hospitals (93 in 2012 and 97 in 2019) in our data that were associated with EINs that included hospitals from different systems. These hospitals were dropped from our analysis, so that we could collapse the compensation data to the system level.”

129. Then hospitals in systems have multiple CEOs. Chose the highest paid CEO? I checked a chain I am familiar with. The head of a hospital that is part of the chain but not the leading hospital has a “President.”

Thank you for letting us know there is not enough detail in the main text for the reader to understand our methodology. Appendix S1.2 explains our methods for identifying the CEO of each system in detail, but we felt outlining the entire process was too cumbersome for the main text. We have revised the text in the manuscript on line 137 that now read, 

“After the possible leaders have been flagged, we designate the “CEO” of each system or independently operated hospital in our sample as the highest paid executive who is flagged as a leader for each organization. A deeper explanation of how we identified CEOs is available in appendix Figs S1.2 and S1.3.[23].”

144. Be more specific about how hospital bed size is defined in the case of systems.

We have corrected the text in our methods section to define how bed size is defined for systems. On line 153, we added the text:

 “We constructed three categorical measures of hospital or system size. For systems in our sample, these measures reflect the total number of beds, adjusted discharges, or hospitals in the system”. 

156. You know from your data that profits are skewed to the right and that there are negative valu

---

## [Decision Letter · Decision Letter 1]

20 Jun 2024

The determinants of nonprofit hospital CEO compensation

PONE-D-24-07430R1

Dear Dr. Jenkins,

We’re pleased to inform you that your manuscript has been judged scientifically suitable for publication and will be formally accepted for publication once it meets all outstanding technical requirements.

Kind regards,

Gabriel A. Picone

Academic Editor

PLOS ONE

Additional Editor Comments (optional):

Reviewers' comments:

Reviewer's Responses to Questions

**Comments to the Author**

1. If the authors have adequately addressed your comments raised in a previous round of review and you feel that this manuscript is now acceptable for publication, you may indicate that here to bypass the “Comments to the Author” section, enter your conflict of interest statement in the “Confidential to Editor” section, and submit your "Accept" recommendation.

Reviewer #1: All comments have been addressed

2. Is the manuscript technically sound, and do the data support the conclusions?

Reviewer #1: Yes

3. Has the statistical analysis been performed appropriately and rigorously? 

Reviewer #1: Yes

4. Have the authors made all data underlying the findings in their manuscript fully available?

Reviewer #1: No

5. Is the manuscript presented in an intelligible fashion and written in standard English?

Reviewer #1: Yes

6. Review Comments to the Author

Reviewer #1: I have read all the comments I had on the previous draft, and my concerns have been addressed. I also read the revised paper.

7. PLOS authors have the option to publish the peer review history of their article (what does this mean?). If published, this will include your full peer review and any attached files.

Reviewer #1: **Yes: **Frank A. Sloan, Ph.D.

---

## [Editor Report · Acceptance letter]

2 Jul 2024

PONE-D-24-07430R1 

PLOS ONE

Dear Dr. Jenkins, 

I'm pleased to inform you that your manuscript has been deemed suitable for publication in PLOS ONE. Congratulations! Your manuscript is now being handed over to our production team.

Kind regards, 

on behalf of

Dr. Gabriel A. Picone 

Academic Editor

PLOS ONE